# Pyroptosis Modulators: New Insights of Gasdermins in Health and Disease

**DOI:** 10.3390/antiox12081551

**Published:** 2023-08-03

**Authors:** Imane Allali-Boumara, Ana Dácil Marrero, Ana R. Quesada, Beatriz Martínez-Poveda, Miguel Ángel Medina

**Affiliations:** 1Andalucía Tech, Departamento de Biología Molecular y Bioquímica, Facultad de Ciencias, Universidad de Málaga, E-29071 Málaga, Spain; imaneall@uma.es (I.A.-B.); anadacil@uma.es (A.D.M.); quesada@uma.es (A.R.Q.); bmpoveda@uma.es (B.M.-P.); 2Instituto de Investigación Biomédica y Plataforma en Nanomedicina-IBIMA Plataforma BIONAND (Biomedical Research Institute of Málaga), E-29071 Málaga, Spain; 3CIBER de Enfermedades Raras (CIBERER), Instituto de Salud Carlos III, E-28029 Madrid, Spain; 4CIBER de Enfermedades Cardiovasculares (CIBERCV), Instituto de Salud Carlos III, E-28029 Madrid, Spain

**Keywords:** non-apoptotic cell death, pyroptosis, gasdermin, natural compounds

## Abstract

Pyroptosis is an inflammation-dependent type of cell death that has been in the spotlight for the scientific community in the last few years. Crucial players in the process of pyroptosis are the members of the gasdermin family of proteins, which have been parallelly studied. Upon induction of pyroptosis, gasdermins suffer from structural changes leading to the formation of pores in the membrane that subsequently cause the release of pro-inflammatory contents. Recently, it has been discovered that oxidation plays a key role in the activation of certain gasdermins. Here, we review the current knowledge on pyroptosis and human gasdermins, focusing on the description of the different members of the family, their molecular structures, and their influence on health and disease directly or non-directly related to inflammation. Noteworthy, we have focused on the existing understanding of the role of this family of proteins in cancer, which could translate into novel promising strategies aimed at benefiting human health. In conclusion, the modulation of pyroptosis and gasdermins by natural and synthetic compounds through different mechanisms, including modification of the redox state of cells, has been proven effective and sets precedents for future therapeutic strategies.

## 1. Introduction

The current understanding of the gasdermin family of proteins is relatively recent, primarily dating back to the last 5 years. However, the first reference describing a member of this family appeared in 1999 in a review on hereditary hearing loss and deafness in which pejvakin (PJVK), a protein currently known as gasdermin F, was mentioned [1]. The term gasdermin appeared for the first time in an article published in 2000, describing the chromosomal location and the expression pattern of gasdermin A in the upper gastrointestinal tract of mice [2]. Interestingly, the first review devoted to the gasdermin family was published (in Chinese) as soon as 2006 [3]. It is noteworthy that a simple search carried out by us in Pubmed showed that out of the 2260 articles published before 30 April 2023 containing the term “gasdermi*”, 2138 had dates of publication from 2018 on (Figure 1). Thus, most of the current knowledge on gasdermins has been published in the last 5 years. It is noteworthy that in just the first four months of 2023 (January to April), over half the number of articles published in the previous year have already been released, underscoring the increasing, keen interest of the scientific community in the field of gasdermins. 

The interest in this family of proteins arises from its implication in pyroptosis, a caspase-dependent cell death that progresses with inflammatory events. Indeed, pyroptosis was defined in 2015 as a gasdermin-mediated programmed death mechanism [4], since these proteins are the effectors of the process [5,6]. Furthermore, recent works have informed about the important role of the redox state of cells in the regulation of this type of inflammatory cell death [7]. The implication of pyroptosis in various diseases, particularly its pivotal role in cancer, has been studied, thereby identifying gasdermins as potential targets of antitumor therapies [8]. The aim of this review is to provide a concise summary of the current knowledge concerning gasdermins and their functions in human health and disease to subsequently review the available evidence regarding synthetic and natural modulators of the process, with a particular focus on their implications in cancer. 

## 2. Structural Aspects of the Gasdermin Family in Humans

While 10 members of the gasdermin family have been identified in mice, only 6 members, known as gasdermins (GSDM) A to F, have been found in humans [9,10]. Interestingly, gasdermins E and F, also called DFNA5 and DFNB59, respectively, have been related to deafness. The chromosomal location and the expression pattern of the proteins have been previously described, showing predominant expression in the gastrointestinal tract and skin (Table 1). 

The gasdermin family members exhibit an average homology of 45% in their sequence, and most of them are characterized by the presence of two highly conserved domains in their structure: the N-terminal (Nt) domain, involved in the formation of a pore-structure, and the C-terminal (Ct) domain, which represses the Nt domain through a common autoinhibitory mechanism [11,12,13]. Indeed, while these two domains are found in GSDMA to GSDME, GSDMF lacks the Ct domain, and its ability to form pores in the membranes has not been verified yet [11,14] (Table 1).

Inflammatory caspases mediate the proteolytic cleavage of gasdermins, releasing the cytotoxic Nt domain from intramolecular inhibition. This enables the Nt domain to interact with membrane phospholipids in the inner leaflet of cell membranes, leading to its oligomerization and the formation of “gasdermin channels” [15]. The mechanism involved in human GSDMD membrane pore formation is well defined [16] and shares characteristics with the activation of the other pore-forming gasdermins [17].

The Nt domain is rich in β sheets, with some ⍺ helixes involved in lipid binding and membrane insertion, whereas the Ct domain only contains ⍺ helixes. In the basal form, the Ct domain folds over the Nt domain, forming electrostatic, hydrophobic, and hydrogen bridges and blocking membrane interaction sites, thus maintaining the whole gasdermin in an autoinhibited state [18,19]. The structures of Ct-containing gasdermins are recapitulated in Figure 2.

**Table 1 antioxidants-12-01551-t001:** Chromosomal localization of the different human gasdermins and their expression pattern.

Human GSDM	Chromosomal Location	Expression Pattern	Pore Formation
GSDMA	17q21	Gastrointestinal tract and skin	Yes
GSDMB	17q21	Cells of the gastrointestinal and respiratory tracts, T cells, and cervix and breast cells	Yes
GSDMC	8q24.2	Cells of the gastrointestinal tract, trachea, and skin	Yes
GSDMD	8q24.2	Cells of the gastrointestinal tract, macrophages, neutrophiles, monocytes, and T and B cells	Yes
GSDME	7p15	Cells of the gastrointestinal tract, placenta, cochlea, and brain	Yes
GSDMF	2q31.1-q31.3	Cells of the inner ear	ND

Information recapitulated from [14]. ND, not determined.

## 3. Functional Features of the Different Members of the Human Gasdermin Family

### 3.1. GSDMA

Gasdermin A was the inaugural family member to be identified when its coding gene was detected by positional cloning in the Rim3 mutant mouse, which exhibited abnormal hair and skin development [2]. Remarkably, the N-terminal functional domain of gasdermin A can target the mitochondrial membranes, thus regulating mitochondrial homeostasis and radical oxygen species formation [20]. It has also been reported that GSDMA gene expression is influenced by asthma-associated polymorphisms in 17q2 [21].

### 3.2. GSDMB

This type of gasdermin exhibits the general characteristics of the family. The documented existence of GSDMB variants with distinct transcript shapes provides compelling evidence for the existence of diverse inhibitory domain mechanisms [22]. It is worth mentioning that for GSDMB to participate in pyroptotic processes, caspase-1 is required to mediate proteolytic cleavage [23].

Some GSDMB variants have been associated with inflammatory diseases such as asthma [17]. The involvement of GSMB in cancer will be further discussed in the Section 5.

### 3.3. GSDMC

Gasdermin C remains relatively underexplored compared to other members of the gasdermin family, although there are experimental data suggesting its potential involvement in certain inflammatory processes. As for other GSDM family members, the overexpression of the N-terminal domain of GSDMC triggers pyroptosis-like features in certain cell types [11]. Interestingly, it has been demonstrated that the N-terminus of gasdermin C induces cytotoxicity, but the mechanisms controlling its activation are still unknown [24]. 

### 3.4. GSDMD

Gasdermin D is the most studied member of the gasdermin family due to its relevant role in inflammasome biology [19], and it is considered a direct substrate of inflammation and a key effector of pyroptosis [13,25,26,27]. GSDMD initiates pyroptosis upon cleavage by inflammatory caspases (1, 4, 5, or 11) [4,25,28]. However, infection of certain pathogens, like *Yersinia*, primes a different caspase (caspase-8) to cleave GSDMD [29]. Interestingly, reactive oxygen species (ROS) promote GSDMD oligomerization through cysteine residue oxidation, which is crucial for GSDMD cleavage and oligomerization to occur [7]. This means that changes in the redox state of the cell would potentially modulate pyroptosis, thereby offering significant therapeutic prospects. After cleavage, the GSDMD N-terminal domain can bind to both the cell membrane and cardiolipin, which is present in the mitochondrial membrane [30]. In addition to its crucial role in pyroptosis, GSDMD is also essential in NETosis progression (a program for formation of neutrophil extracellular traps (NETs)) [31,32]. 

GSDMD is an important mediator in the host response to Gram-negative bacteria [25]. Remarkably, GSDMD-mediated pyroptosis is driven by lipid peroxidation in cases of lethal polymicrobial sepsis [33]. Conversely, inhibition of GSDMD could prevent inflammatory cell death and sepsis, as has been demonstrated in the case of necrosulfonamide, a direct chemical inhibitor of this gasdermin [34]. These findings suggest that GSDMD inhibitors could be used in targeted therapies for patients suffering from sepsis [35]. The overall roles of GSDMD in inflammation and host defense have been extensively reviewed by Lieberman et al. [36]

It is noteworthy that despite the pro-inflammatory effects of GSDMD, this protein can also exert anti-inflammatory effects by inducing lytic cell death in neutrophils [37].

GSDMD-mediated pyroptosis has been related to several diseases and pathological states, including acute kidney injury [38], ischemic stroke [39], and cancer (see Section 5). Hence, the motivation to pursue drug discovery targeting GSDMD is evident. In this regard, disulfiram, a drug used to treat alcohol addiction, was able to block GSDMD-mediated pore formation, therefore inhibiting pyroptosis [40]. In addition, there is evidence that emodin, an anthraquinone derivative from the rhizome of *Rheum palmatum*, can alleviate certain cardiac lesions produced as a consequence of pyroptosis triggered by GSDMD in an animal model [41].

### 3.5. GSDME

GSDME is considered to be another effector of pyroptosis, although in a lesser proportion than GSDMD [27]. This gasdermin has also been related to cases of secondary necrosis in response to apoptotic stimuli and hearing loss, among other pathologies [14]. Notably, cleaved GSDME pores can permeabilize not only the plasma membrane but also the inner mitochondrial membrane, leading to cytochrome c release from the mitochondria and, eventually, inflammasome activation [42].

### 3.6. GSDMF

Gasdermin F, also known as pejvakin and DFNB59, stands out as a structural anomaly within the gasdermin family because it lacks the regulatory Ct domain, suggesting a potential evolutionary divergence [19].

This type of gasdermin is related to progressive deafness due to an autosomal recessive mutation [17]. Moreover, pejvakin mediates the autophagic degradation of peroxisomes, a process also referred to as pexophagy. Notably, pexophagy has been demonstrated to be a defense mechanism that safeguards auditory hair cells against damage induced by sound overstimulation [43].

## 4. Involvement of Gasdermin Pore Formation in Pyroptosis

Originally, in 1992, the term pyroptosis was described as a form of apoptosis occurring during infection by the Gram-negative bacterium *Shigella flexneri*. Subsequently, in 2001, the pyroptotic process came to be appreciated and classified as a different type of cell death with an inflammatory nature [24]. The Nomenclature Committee on Cell Death (NCCD) recently redefined pyroptosis as a type of programmed cell death mediated by the formation of GSDM pores in the plasma membrane usually triggered by the activation of inflammatory caspases [44]. 

Pyroptosis occurs in response to pathogenic infections or cell damage, whereby lysis of affected immune cells occurs following GSDM-mediated pore formation. These pores facilitate the massive release of cellular contents, which will finally trigger inflammation [11]. The inflammatory response may have a protective role, but, when uncontrolled, it has been related to pathologies such as inflammatory bowel disease or gout [35]. Indeed, an imbalanced pyroptosis response can lead to severe issues, impacting the immune system and triggering the so-called “septic shock” [19].

The process of pyroptosis occurs as follows. After the presence of a threat, inflammasomes are activated [45]. These inflammasomes are responsible for the initiation of the cascade of pro-inflammatory caspases [19] associated with the maturation and release of interleukins, among which IL-1β acts as a substrate of caspase-1 [45]. Briefly, the main processes that occur for GSDMD pore formation are: (1) Caspase-1 activation through the inflammasome upon detection of a potential threat. (2) GSDMD cleavage triggered by oxidation of specific residues and release of the Nt-GSDMD domain. (3) Processing and maturation of IL-1β. (4) Oligomerization of the GSDMD N-t domain at the plasma membrane. (5) Formation of the GSDMD pore. (6) Release of Il-1β. A scheme of the process is provided in Figure 3.

Depending on the type of stimulus or threat, either the canonical inflammasome or the non-canonical inflammasome will be activated [46]. Canonical inflammasomes recognize signals such as bacterial flagellin, K^+^ efflux, or cytosolic DNA that activate caspase-1. Conversely, non-canonical inflammasomes recognize bacterial lipopolysaccharides (LPSs), present in the outer membrane of Gram-negative bacteria, or oxidized lipids, which activate caspase-11, caspase-4, or caspase-5 [22]. Regardless of the type of activation pathway, membrane pore formation and interleukin release will be achieved due to GSDMD cleavage. GSDMD pore formation has also been linked to IL-33 release [47].

The connections of GSDMD (and other members of the gasdermin family) with inflammation and cell death have been comprehensively reviewed elsewhere [5,9,17]. In the context of gasdermins and inflammation, GSDME-mediated pyroptosis has been shown to promote intestinal inflammation, contributing to the pathogenesis of Crohn’s disease [48]. The identification of chemicals targeting the overall process is also noteworthy [19]. This is the case of the afore-mentioned inhibitory effect of disulfiram-blocking GSDMD pore formation [40], the modifying effect of the immunomodulator metabolite itaconate on NLRP3 inflammasome [49,50,51], and the inactivation of GSDMD by succinylation elicited by dimethyl fumarate [52]. Other natural compounds inhibiting inflammation and pyroptosis via the interference of gasdermin function include emodin, an anthraquinone [41]; dimethyl fumarate [51,53]; honokiol, a lignan [54]; and oridonin, a diterpenoid [55]. Remarkably, quercetin, a natural antioxidant [56], protects hepatocytes and macrophages from pyroptosis via ROS scavenging and NRF2 activation, respectively [57,58]. Interestingly, these two mechanisms involved in the inhibitory action of this antioxidant on cell pyroptosis are related to the oxidation-dependent activation of GSDMD.

The role of pyroptosis (and gasdermins) in different diseases has been extensively reviewed elsewhere [9]. For instance, pyroptosis has been proved to be important in the development of cardiovascular disease [59,60,61,62,63], liver diseases [64,65], cancer [8,66,67,68,69], and central nervous system disorders [70,71,72]. Intriguingly, pyroptosis is also involved in the pathogenesis of atherosclerosis [73,74]. It has also been claimed that targeting GSDMD could be a strategy for ischemic stroke therapy [75]. In this context, the fact that hydroxytyrosol acetate, a phenolic compound found in leaves, olives, and olive oil, inhibits pyroptosis in vascular endothelial cells, offers new insight into the described cardioprotective effects of hydroxytyrosol [76]. The contributions of pyroptosis in kidney diseases [77,78], COVID-19 [79,80,81], and diabetes [82] have also been studied and reviewed elsewhere. Moreover, the connection of pyroptosis with inflammatory diseases is clearly established and has been extensively reviewed elsewhere [83,84], including inflammation-related respiratory disease [85,86], inflammatory bowel disease [87,88], and rheumatoid arthritis [89]. 

## 5. Gasdermins, Pyroptosis, and Cancer

As soon as 2000, it was claimed that gasdermins are frequently suppressed in gastric cancer [2]. Since then, evidence has accumulated connecting gasdermins and pyroptosis with cancer [90,91,92,93,94,95]. The immunogenic nature of pyroptosis endows it with great potential to exert antitumor effects. However, the activation of pyroptosis and different types of gasdermins in specific cancer cell types may be different, leading to diverse outcomes depending on the cancer background. The impact of pyroptosis on cancer development is multifaceted, including the suppression of cancer cell viability, modulation of cancer cell invasion and migration, reinforcement of antitumor immune responses, and increase of chemotherapy sensitivity [95]. Therefore, modulation of pyroptosis in a controlled manner, under appropriate conditions, can be exploited as a valuable strategy in cancer treatment. In this section, we review the evidence regarding the role of pyroptosis and the members of the gasdermin family in different type of tumors and cancer cells, as well as synthetic and natural molecules that have been shown to have an effect on this mechanism of inflammatory cell death, highlighting the role of oxidation. 

Natural and synthetic compounds that activate pyroptosis specifically in cancer cells could be candidates for cancer chemoprevention or therapy [96,97,98] (Table 2). This seems to be the case of a natural product from *Vernonia cinerea* (a plant belonging to the Asteraceae family), which selectively induces pyroptosis in triple-negative breast cancer (TNBC) cells compared to normal human breast epithelial cells. This induction of pyroptosis in TNBC cells is preceded by a reduction of glutathione (GSH) levels and an increase of ROS [99]. In addition, anthocyanin, a flavonoid, has been shown to activate pyroptosis in oral squamous cell carcinoma [100]. Galangin, another flavonoid, can induce pyroptosis, along with apoptosis and autophagy, in glioblastoma [101]. The *N,N*-dimethylbiguanide metformin, a first-line drug for the treatment of type 2 diabetes mellitus, has demonstrated capacity to induce pyroptosis in human esophageal carcinoma cells [102]. 

Moreover, combined BRAF and MEK inhibitors have been approved by the US Food and Drug Administration (FDA) to treat melanoma patients with V600E/K mutation in the *BRAF* gene. A recent study has shown that such a combination of inhibitors induces alterations in the tumor immune microenvironment associated with an increased cancer cell death via pyroptosis [103]. In this line, newly synthetized phenylsulfonimide PPAR⍺ antagonists induced apoptosis and pyroptosis of MCF7 breast cancer cells via Nrf2 activation and promotion of oxidative stress [104]. 

Remarkably, GSDMB overexpression has been associated with several types of cancer [23]. In fact, it has been reported that this gasdermin promotes both invasion and metastasis in breast cancer [12]. In patients with human epidermal growth factor receptor 2 positive (HER2+) breast cancer, high GSDMB expression levels correlate with poor prognosis and poor therapeutic response [105]. Conversely, an antibody-blocking GSDMB can reduce HER2+ breast cancer aggressiveness [106]. Therefore, this type of gasdermin is considered to promote invasion and metastasis in this type of cancer, suggesting that it might be a good marker for the stratification of these patients [12,23,105]. 

In general, GSDMC upregulation in cancer correlates with poor survival [91]. Certainly, the downregulation of this gasdermin reduces colorectal cancer cell line proliferation and, reversely, its upregulation promotes colorectal cancer proliferation [107]. In the case of lung adenocarcinoma, GSDMC expression is a prognostic factor of a poor outcome [108]. Under hypoxia, phospho-serine acetyltransferase 3 (p-Sat3) interaction with programmed death-ligand 1 (PD-L1) triggers its nuclear translocation, thus enhancing *GSDMC* transcription. This observation points to a non-immune checkpoint function of PD-L1, switching apoptosis to pyroptosis and facilitating tumor necrosis [109]. In addition, an interesting report shows that treatment with dimethyl ⍺-ketoglutarate, a cell-membrane-permeable precursor of ⍺-ketoglutarate, induces GSDMC-dependent pyroptosis in HeLa cells [110].

GSDMD downregulation promotes gastric cancer proliferation through ERK activation and acceleration of the S/G_2_ transition in cell cycle [111]. In high contrast, it produces the opposite effect in non-small cell lung cancer, where it can be considered a predictor of good prognosis [112]. Similarly, the overexpression of GSDMD relates to the promotion of invasion in the case of adenoid cystic carcinoma [113]. Using The Cancer Genome Atlas (TCGA) cohorts, it has been demonstrated that GSDMD gene expression consistently correlates with markers for CD8^+^ T cells and that GSDMD is required for an optimal response of cytotoxic T lymphocytes to cancer cells [114]. In contrast, a recent publication proved that GSDMD is highly expressed in antigen-presenting cells in tumors, and that inhibitors of this gasdermin improve the antitumor immunity during anti-PD-L1 treatment [115]. Treatment with trichosanthin, a protein extracted from the root tuber of *Trichosanthes kirilowii*, significantly increases the expression of GSDMD and NLRP3 in A549 non-small cell lung cancer cells, thus inhibiting their proliferative and metastatic potential [116]. On the other hand, it has been shown that secoisolariciresinol diglucoside, a lignan isolated from flaxseed, induces GSDMD-mediated pyroptosis in colorectal cancer cells [117]. 

A tumor-growth-suppressor function has been proposed for GSDME, based on the observation that a decrease in the expression of this gasdermin correlates, in certain cancers, with a decreased survival, as it activates antitumor immunity. This effect of GSDME is noticeable in the enhancement of tumor cell phagocytosis by macrophages [118]. In addition, it has been observed that the N-terminal domain of GSDME, after its cleavage mediated by caspase-3, triggers the intrinsic apoptotic pathway of cancer cells [42]. Furthermore, there are reports suggesting that chemotherapy drugs can trigger pyroptosis through the activation of caspase-3 and cleavage of GSDME. This mechanism has the potential to pave the way for novel approaches in cancer chemotherapy [119]. In this regard, GSDME increases chemotherapeutic drug sensitivity in retinoblastoma cells by inducing pyroptosis [120]. Interestingly, the bioactive phenolic compound curcumin induces pyroptosis on hepatocellular carcinoma HepG2 cells via activation of ROS signaling, which increases the expression of the GSDME N-terminal domain [121]. The sesquiterpene germacrone also induces HepG2 pyroptosis via GSDME through an increase of ROS production [122]. Similarly, aloe-emodin, an anthraquinone, activates the caspase 3/9-GSDME axis in HeLa cells, giving rise to mitochondrial dysfunction and pyroptosis [123]. It has been suggested that GSDME can be used simultaneously as a pan-cancer biomarker and as a marker able to discriminate between different types of cancer [124]. It is noteworthy that the caspase-3/GSDME signaling pathway can behave as a switch between pyroptosis and apoptosis in cancer [125]. In fact, dihydroartemisinin, a semi-synthetic derivative of artemisin, a natural lactone, induces pyroptosis in breast cancer cells through promotion of the caspase-3/GSDME axis [126]. The selective activator of retinoid X receptor bexarotene induces caspase-4-GSDME mediated pyroptosis in ovarian cancer cells [127]. Recently, it has been reported that simvastatin in combination with a DNA methyltransferase inhibitor induces GSDME-mediated pyroptosis, resulting in a relevant antitumor effect [128]. Tumorigenesis in colitis-associated colorectal cancer is promoted by GSDME-mediated pyroptosis, which leads to the release of high mobility group box 1 proteins (HMGB1) [129]. The roles of GSDME in cancer have been recently reviewed [124,130,131,132]. Interestingly, wedelolactone, a polyphenol isolated from *Wedelia chinensis* and *Eclipta alba,* simultaneously activates GSDMD and GSDME and increases caspases-1 and -3, thus inducing pyroptosis and apoptosis in retinoblastoma [125].

**Table 2 antioxidants-12-01551-t002:** Natural and synthetic compounds that induce pyroptosis.

Compound	Type of Cancer	Pyroptosis-Related Mechanism	Experimental Models	Reference
**Aloe-emodin**	-	Increased pyroptosis via activation of the caspase 3/9-GSDME axis in HeLa cells	HeLa cells	[123]
**Anthocyanin**	Oral squamous cell carcinoma (OSCC)	Activation of pyroptosis through enhanced expression of NLRP3, caspase-1, and IL-1β	HaCaT, Tca8113, and SCC15 cells	[100]
**Bexarotene**	Ovarian cancer (OC)	Induction of Caspase-4-gasdermin E mediated pyroptosis	ES2 and NIH:OVCAR3 OC cell lines.	[127]
**Combination of BRAF inhibitors and MEK inhibitors (BRAFi + MEKi)**	*BRAF* ^V600E/K^-mutant melanoma	Increased GSDME-mediated pyroptosis related to ERK1/2 pathway inhibition	Human and syngeneic mouse melanoma models	[103]
**Curcumin**	Hepatocellular Carcinoma	Augmented pyroptosis through increased expression of GSDME N-terminal domain	HepG2 cells	[121]
**Dihydroartemisinin**	Breast cancer (BC)	Induced pyroptosis through promotion of the caspase-3/GSDME axis	MCF-7 and MDA-MB-231 cell lines. BC xenografts in BALB/c nude mice	[126]
**Dimethyl** **⍺** **-ketoglutarate**	-	Induced GSDMC-dependent pyroptosis through death receptor 6-activated caspase-8	HeLa and other cell lines. Male nude mice and C57BL/6J mice	[110]
**Galangin**	Human glioblastoma multiforme (GBM)	Simultaneous induction of apoptosis, pytoptosis, and autophagy	U251, U87MG, and A172 cells. Orthotopic xenograft model in nude mice derived from U87MG cells	[101]
**Germacrone**	Liver cancer	Induced GSDME-dependent pyroptosis through caspase 3 activation	HepG2 cells. HepG2 cell xenograft models	[122]
***N,N*-dimethylbiguanide metformin**	Esophageal squamous cell carcinoma (ESCC)	Induction of gasdermin D (GSDMD)-mediated pyroptosis by targeting miR-497/PELP1 axis	Tumor samples of human primary ESCC	[102]
**Phenylsulfonimide PPAR** **⍺** **antagonists**	Breast cancer (BC)	Induction of apoptosis and pyroptosis via Nrf2 activation and promotion of oxidative stress	MCF7 cell line	[104]
**Secoisolariciresinol diglucoside**	Colorectal cancer (CC)	GSDMD-dependent pyroptosis through generation of ROS/P13K/AKT/BAK-mitochondrial pathway apoptosis	HCT116 cells. HCT116-CRC xenograft model in nude mice	[117]
**Simvastatin + DNA methyltransferase inhibitor**	Gastric Cancer (GC)	Increased GSDME-mediated pyroptosis	Human samples of GC tissue. GC cell lines AGS, MKN45, HGC-27, and MGC-803. GC xenograft model in mice.	[128]
**Trichosanthin**	Non-small cell lung cancer (NSCLC)	Increased pyroptosis via expression of GSDMD and NLRP3	NSCLC cells. Tumor xenograft model in nude mice derived from A549 cells	[116]
**Wedelolactone**	Retinoblastoma (RB)	Pyroptosis and apoptosis induction via simultaneous activation of GSDMD and GSDME, and caspases 1 and 3	RB cell lines Y79 and Weri-Rb1. Xenograft experiments on BALB/c nude mice derived from Y79 cells.	[125]

## 6. Concluding Remarks

Among the mechanisms of cell death different from apoptosis, pyroptosis comes as a significant, inflammation-related pathway that has exponentially grabbed the attention of the scientific community in the last few years. Since the first reference to gasdermins as proteins associated with hereditary hearing loss and deafness [1], the different members of the family have become increasingly known. Furthermore, their molecular structures, their role in pyroptosis, and their influence on health and disease have been described. In this study, we have reviewed the current knowledge on pyroptosis and human gasdermins, focusing on the above-mentioned aspects. 

Interestingly, the self-inhibitory structure of the members of the gasdermin family allows for their cleavage by specific caspases in response to a pro-inflammatory stimulus, with a consequent oxidation-triggered oligomerization of the N-terminal domain in the plasma or mitochondrial membrane upon formation of a pore and release of content, prompting the inflammatory response. This inflammation-related feature of pyroptosis makes it an important process in the host-microbe defense and the response to other detrimental stimuli but, conversely, it also contributes to the pathogenesis of different diseases, such as Crohn’s disease, diabetes, COVID-19, or certain types of cancer [9,48,82,93]. Thus, modulation of pyroptosis has gained great interest in pharmacological research in the last few years as it presents an opportunity for new therapeutic approaches. In this line and provided the redox-related nature of GSDMD (the most important member of the gasdermin family) oligomerization for pore formation, this inflammatory cell death is potentially altered by modulation of the redox state of cells.

Noteworthy, the fact that pyroptosis is repressed in many types of cancers suggests that synthetic and natural compounds able to activate this process could be used in novel and valuable therapeutic options. Interestingly, studies have shown that there is a crosstalk between anticancer immunity and nonapoptotic cell death mechanisms [133]. In this line, lab-designed products like anti-GSDMB antibodies or phenylsulfonimide PPAR⍺ antagonists; and natural compounds like anthocyanin, curcumin, dimethyl-α ketoglutarate, or metformin, have shown remarkable pro-pyroptotic effects in diverse in vitro and in vivo studies (see Table 2). However, there is only one currently active clinical trial involving the study of an agent on endothelial pyroptosis by targeting GSDMDs, and this trial is not cancer-related [134]. Thus, we encourage the scientific community to increase the knowledge of the mechanisms of gasdermins and pyroptosis and to consider these modulatory molecules for future synergistic studies with other approaches like traditional chemotherapy or the emerging modulation of cancer immunity.

## Figures and Tables

**Figure 1 antioxidants-12-01551-f001:**
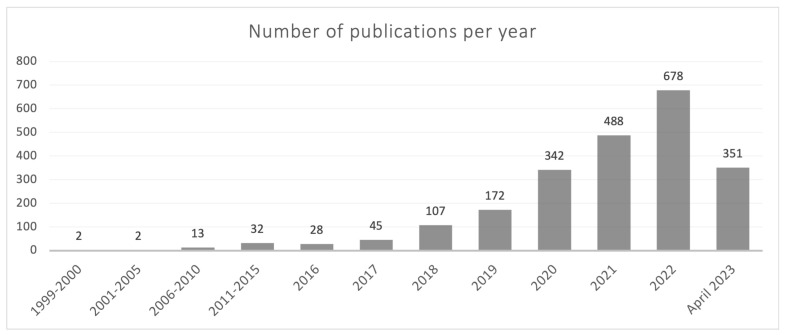
Year-distribution profile of publications in Pubmed containing the term “gasdermi*” from 1999 to 30 April 2023.

**Figure 2 antioxidants-12-01551-f002:**
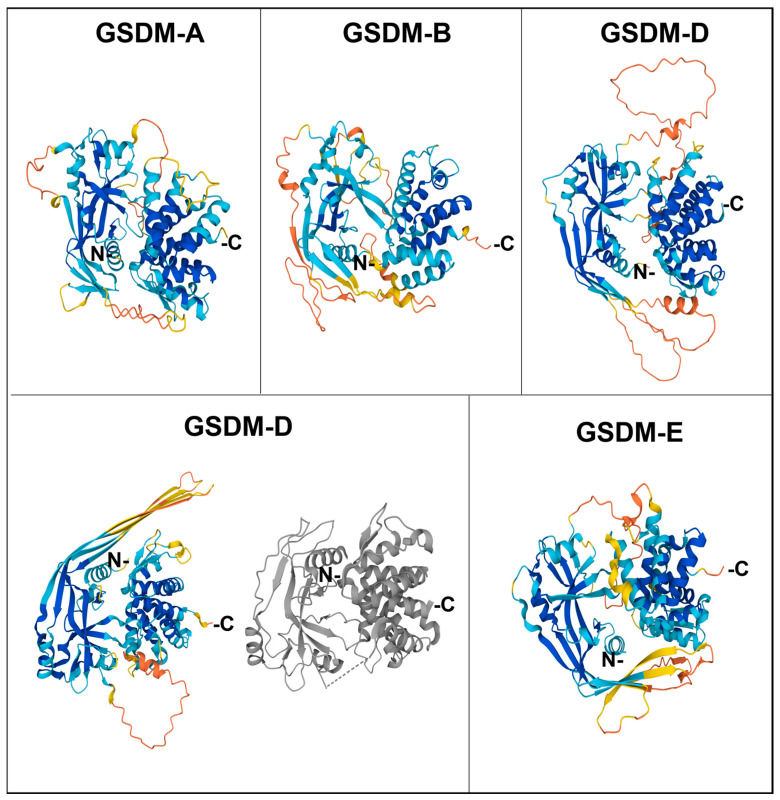
Structure of human gasdermins. The colored figures correspond to the predicted structures by AlphaFold, with the following model of confidence: blue, Very high (pLDDT > 90); cyan, Confident (90 > pLDDT > 70); yellow, Low (70 > pLDDT > 50), and orange, Very low (pLDDT < 50). For gasdermin D, the most studied one, the structure of the crystallized protein from PDB is also shown (in grey).

**Figure 3 antioxidants-12-01551-f003:**
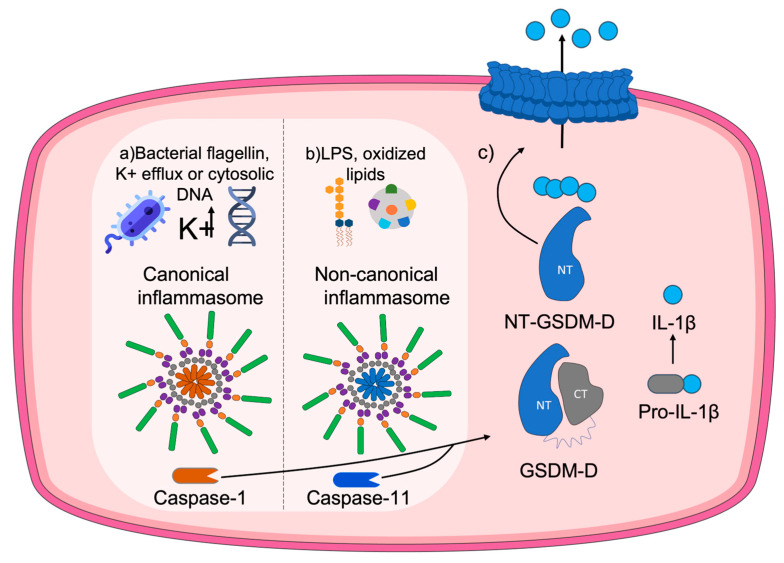
Molecular mechanism of the pore formation by GSDMD. When the canonical pathway is activated (**a**) by stimuli such as bacterial flagellin, potassium efflux, or cytosolic DNA, the canonical inflammasome forms, promoting cleavage of pro-caspase-1 into the inflammatory caspase-1. A similar procedure occurs throughout the non-canonical pathway (**b**), induced by stimuli like LPS from Gram-negative bacteria or oxidized lipids and involving caspase-11 instead. Both pathways converge in (**c**) the gasdermin D (GSDMD) oxidation-dependent cleavage and release of the N-terminal domain, together with the maduration of IL-1β. Finally, the N-terminal domain of GSDMD is transported to the membrane, where it oligomerizes with other N-terminal domains to eventually form a pore in the plasma membrane that leads to the release of pro-inflammatory content to the extracellular media.

## Data Availability

Not applicable.

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
