# Peer review of "Pyroptosis Modulators: New Insights of Gasdermins in Health and Disease"

_antioxidants, 2023, doi:10.3390/antiox12081551_

Round 1

Reviewer 1 Report

The authors have reviwed the reserch development on pyroptosis and human gasdermins, focused on different members, molecular structures and influence on heath and disease. The review article is well written with proper references. 

The  role of oxidative stress can be elaborated with the recent references. 

Antioxidant play an imortant role also need to be discussed in context of correcting antioxidants and oxidative stress. 

Figures and tables are summarised well. 

Moderate English/spell check required.  

Author Response

The authors want to thank the reviewer or his/her positive comments on our manuscript

Reviewer 2 Report

This paper reviewed the current knowledge on pyroptosis and human gasdermins, focusing on the description of the different members of the family, their molecular structures, and their influence on health and disease directly or non-directly related to inflammation. The authors focused on the existing understanding of the role of this family of proteins in cancer, what could translate into novel promising strategies aimed at the benefit of human health. The modulation of pyroptosis and gasdermins by natural and synthetic compounds through different mechanism, like modification of the redox state of cells, has been proven effective and sets precedents for future therapeutical interventions.

  1. Though the study is quite interesting, the reviewed aspects of pyroptosis is kind of narrow. The authors should discuss if there is any other similar research work.
  2. The overall writing has some formatting issues, like wording and spacing. I suggest the authors check the grammar and avoid any typos. More importantly, the writing needs improvement.
  3. I would suggest the authors to highlight the roles of pyroptosis in a specific cancer type, which helps enhance the depths of this review.

N/A

Reviewer 3 Report

The manuscript by Allali-Boumara et al. describes the different members of the family of gasdermins and aims to review the role of this family of proteins in pyroptosis and cancer. The topic is very interesting,  and different aspects have been reviewed  in the last years.  The manuscript is clear and well written and provides a table that lists the natural and synthetic compounds that are known to be able to modulate pyroptosis and gasdermins, and the mechanisms involves.  There are some minor issues that should be corrected:

Line 118 “Yersinia” and Line 144 “Rheum palmatum” should be written in italics.

168 Ref. 19 does not seem to be adequate for the sentence: “The Nomenclature Committee on Cell Death  (NCCD), recently redefined pyroptosis as a type of programmed cell death mediated by  the formation of GSDM pores in the plasma membrane usually triggered by the activation  of inflammatory caspases” [19].

Please consider Galluzzi L, et al. Molecular mechanisms of cell death: recommendations of the Nomenclature Committee on Cell Death 2018. Cell Death Differ. 2018 Mar;25(3):486-541. doi: 10.1038/s41418-017-0012-4.

Table 2: Dimethyl a-ketoglutarate: a should be substituted with alpha.

Line 214 succination: succinylation

Author Response

Reviewer 3

The manuscript by Allali-Boumara et al. describes the different members of the family of gasdermins and aims to review the role of this family of proteins in pyroptosis and cancer. The topic is very interesting,  and different aspects have been reviewed  in the last years.  The manuscript is clear and well written and provides a table that lists the natural and synthetic compounds that are known to be able to modulate pyroptosis and gasdermins, and the mechanisms involves.  

WE THE AUTHORS WANT TO THANK THE REVIEWER FOR HIS/HER COMMENTS AND SUGGESTIONS.

There are some minor issues that should be corrected:

Line 118 “Yersinia” and Line 144 “Rheum palmatum” should be written in italics.

Thank you, this has been corrected.

168 Ref. 19 does not seem to be adequate for the sentence: “The Nomenclature Committee on Cell Death  (NCCD), recently redefined pyroptosis as a type of programmed cell death mediated by  the formation of GSDM pores in the plasma membrane usually triggered by the activation  of inflammatory caspases” [19].

Please consider Galluzzi L, et al. Molecular mechanisms of cell death: recommendations of the Nomenclature Committee on Cell Death 2018. Cell Death Differ. 2018 Mar;25(3):486-541. doi: 10.1038/s41418-017-0012-4.

Thank you, we agree with your suggestion and the reference has been changed to the one you propose.

Table 2: Dimethyl a-ketoglutarate: a should be substituted with alpha.

This has been corrected, thank you.

Line 214 succination: succinylation

This change has been made. Thanks again.

Round 2

Reviewer 1 Report

The authors have written a good review but haven't included or responded to reviewer suggestions. 

Minor spell check.

Author Response

Reviewer: The authors have written a good review but haven't included or responded to reviewer suggestions.

Authors: We do not know what could happen in our previous resubmission. We were confident we were properly uploaded all our responses to reviewer's suggestions. This should have been as follows:

The authors have reviwed the reserch development on pyroptosis and human gasdermins, focused on different members, molecular structures and influence on heath and disease. The review article is well written with proper references.

The authors want to thank the very good overall evaluation of our manuscript made by the reviewer.

The role of oxidative stress can be elaborated with the recent references.

Antioxidant play an imortant role also need to be discussed in context of correcting antioxidants and oxidative stress.

In the present, amended version of our manuscript, the authors have tried to emphasize the role of oxidative stress and antioxidants in this issue. In fact, this was an item also commented by the editor int he first round.

Figures and tables are summarised well.

The authors want to thank these positive comments made by the reviewer.

Minor spell check.

The authors have carefully revised the whole manuscript trying to detect and eliminate minor spell and grammar mistakes.

Reviewer 2 Report

Though the authors partially addressed my concerns, this manuscript has moderate scientific impact. 

Quality of English language still needs improvement. 

Author Response

REVIEWER: Though the authors partially addressed my concerns, this manuscript has moderate scientific impact. 

Authors: We thank the reviewer for acknowledging that we have (partially) addressed his/her concerns, as raised in the previous round of revision. For this reason, it is a surprise for us that in the "Review Report Form" the reviewer has diminished the evalution of the five field included in the Report Form from THREE stars in the firs round to TWO stars in the second round. Since the reviewer acknowledges that we have partially addressed his/her concerns, what we. could expect is that he/she would have at least partially increased some of the scores in the Review Report Form.

Quality of English language still needs improvement. 

Author have reviewed thoroughly the whole manuscript, checking and correcting misspelling. 

Round 3

Reviewer 2 Report

I no longer have concerns on this manuscript.